# Zero-shot Image Classification with Logic Adapter and Rule Prompt

Submission Id: 1420

## ABSTRACT

Zero-shot image classification, which aims to predict unseen classes whose samples have never appeared during the training phase, is crucial in the Web domain because many new web images appear on various websites. Attributes, as annotations for class-level characteristics, are widely used semantic information for zero-shot image classification. However, most current methods often fail to capture discriminative image features between similar images from different classes, leading to unsatisfactory zero-shot image classification results. This is because they solely focus on limited semantic alignments between visual and attribute features. Therefore, we propose a **Z**ero-**S**hot image **C**lassification with **L**ogic adapter and **R**ule prompt method called ZSCLR, which utilizes logic adapter and rule prompts to encourage the model to capture discriminative image features and achieve reasoning. Specifically, ZSCLR consists of a visual perception module and a logic adapter. The visual perception module extracts basic image features from training data. At the same time, the logic adapter utilizes the Markov logic network to encode the extracted basic image features and rule prompts for refining the discriminative image features. Due to predicates of rule prompts representing symbolic discriminative features, the proposed model can focus more on these discriminative features and achieve more precise image classification. Additionally, the logic adapter enables the model to adapt from recognizing images in seen classes to those in unseen classes through the reasoning of the Markov logic networks. We implement experiments on two standard zero-shot image classification benchmarks, and ZSCLR achieves competitive performance. Furthermore, ZSCLR can provide explanations for its predictions through rule prompts.

## CCS CONCEPTS

• **Computing methodologies** → **Object recognition**; **Computer vision**; **Semantic networks**.

## KEYWORDS

Zero-shot Learning, Image Classification, Logic Adapter, Rule Prompt, Markov Logic Network

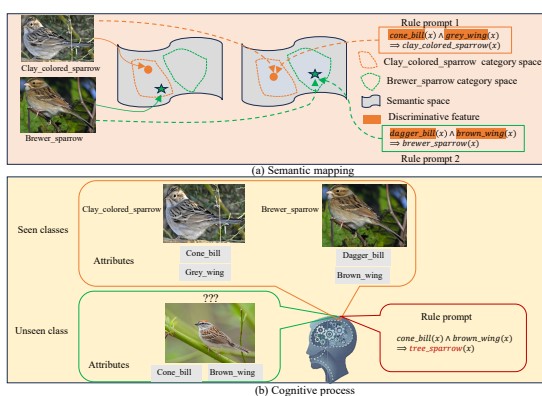

**Figure 1: The illustrative diagrams. (a) Due to the object's visual features being similar, the *Brewer_sparrow* is misclassified as a *Clay_colored_sparrow* in semantic space. It is correctly recognized using discriminative features in the rule prompts by our proposed ZSCLR. (b) The cognitive process of the human for recognizing objects unseen previously.**

ACM Reference Format:
Anonymous Author(s). 2018. Zero-shot Image Classification with Logic Adapter and Rule Prompt. In *Proceedings of ACM Conference (Conference'17)*. ACM, New York, NY, USA, 9 pages. https://doi.org/XXXXXXX.XXXXXXX

## 1 INTRODUCTION

Relying on massive labeled training data to classify images has led to significant progress in the computer vision domain in recent years [22]. However, annotating all objects' classes is unrealistic, especially those that are expensive to label or not efficiently collectible in real-world applications. This limitation poses challenges for supervised learning methods when classifying objects that were unseen during training. To address this issue, zero-shot learning (ZSL) has emerged as a promising approach [15, 35]. ZSL requires labeled objects from seen classes but can effectively recognize objects from unseen classes. Applying ZSL to the computer vision domain is crucial because a vast number of new web images are created daily on social media and other websites.

In this paper, we focus on zero-shot image classification (ZSC), which involves training the model on images of seen categories and recognizing images of unseen categories. Most methods map images from the visual space into the semantic space and classify them [12, 14, 19, 30]. However, these methods may lead to misclassification due to the similarity between image features. For instance, as shown in Figure 1 (a) left, images of instance "*clay_colored_sparrow*" and "*brewer_sparrow*" are mapped into the semantic space, where the image feature of "*brewer_sparrow*" falls into the "*clay_colored_sparrow*" category space since they are visually similar. Based on this situation, we observe that objects with similar appearances exhibit similar image features in the

semantic space, producing a wrong classification. To avoid such misclassifications, the model needs to capture discriminative image features, such as vital attributes that distinguish differences. For example, the discriminative features of "*clay_colored_sparrow*" and "*brewer_sparrow*" are the shape of the bill, e.g., "*cone_bill*" and "*dagger_bill*" and the color of the wing, e.g., "*grey_wing*" and "*brown_wing*". If informing the model about these discriminative features via rule prompts such as cone_bill$(x)\wedge$grey_wing$(x) \Rightarrow$ clay_colored_sparrow$(x)$ and dagger_bill$(x)\wedge$brown_wing$(x)$ $\Rightarrow$ brewer_sparrow$(x)$, images can be better classified in the semantic space, as depicted in Figure 1 (a) right. Therefore, the proper discovery of discriminative image features in semantic spaces for ZSC is of great importance.

Recently, several methods have emerged for learning discriminative image features in ZSC. Researchers have focused more on attention networks, leading to the development of attention-based ZSC methods [3, 14, 36]. These methods leverage attribute descriptions, e.g., word vectors, as auxiliary information to discover discriminative image features, facilitating accurate alignment with semantic representations. Although these efforts improve the classification accuracy in ZSC, the results are still unsatisfactory, particularly when handling datasets containing very visually similar images belonging to different classes. Because these approaches primarily rely on attention networks, which focus on limited semantic alignments between visual and attribute features. Logic rules, which condense human intelligence and knowledge, can be utilized to guide extracting discriminative image features. However, there is currently no existing method to apply them to ZSC tasks in terms of image classification from unseen classes due to the challenge of integrating two distinct representations: logic rules in symbolic form and image representations in vector/matrix form. Therefore, in this paper, we seek to address the following question: *How do we properly use logic rules to capture discriminative image features for improving zero-shot image classification accuracy?*

To address the above issue, we will introduce a logic adapter for integrating logic rules to ZSC for classifying images from unseen classes inspired by the human cognitive process. Humans possess the ability to comprehend unseen objects through reasoning based on prior experiences, even without prior exposure to them. For example, in Figure 1 (b), seen classes are "*clay_colored_sparrow*" with attribute features "*cone_bill*", "*grey_wing*" and "*brewer_sparrow*" with "*dagger_bill*", "*brown_wing*", while unseen class is "*tree_sparr ow*". When humans encounter a new image of a "*tree_sparrow*" for the first time, they rely on prior knowledge from having seen images of "*clay_colored_sparrow*" and "*brewer_sparrow*". By comparing the new image's visual attributes, such as "*cone_bill*" and "*brown_wing*", to what they have learned from seen classes, humans can select the proper rule prompt, i.e., cone_bill$(x)\wedge$brown_wing$(x) \Rightarrow$ tree_sparrow$(x)$, from the knowledge base containing various rule prompts. Thus, they can logically deduce that the new image from the unseen class is indeed "*tree_sparrow*".

Based on the above analysis, we have proposed a method called ZSCLR (**Z**ero-**S**hot image **C**lassification with **L**ogic adapter and **R**ule prompt), which integrates rule prompts into ZSC models to capture discriminative image features. ZSCLR comprises two key components: a visual perception module and a logic adapter. The visual perception module, designed using CNN and attention

network, primarily focuses on basic image feature extraction. Meanwhile, the logic adapter takes basic image features from the visual perception module and encodes them and rule prompts that predicates represent symbolic discriminative features via Markov logic network [16]. More concretely, in the logic adapter, we introduce a statistical relation learning model, i.e., Markov logic network (MLN). MLN can combine statistical models (e.g., ZSC models) and relational models (e.g., rule prompts) to attain a unified representation, such as a joint probability distribution, and achieve reasoning via computing posterior. Moreover, rule prompts with symbolic discriminative features can tell the model which features are discriminative in an image, such as the shape of the bill, and then the model can focus more on what during training. In this paper, rule prompts are formalized through first-order logics (FOLs), such as dagger_bill$(x)\wedge$brown_wing$(x) \Rightarrow$ brewer_sparrow$(x)$, encoding them within the MLN. FOLs serve a dual purpose: they represent logical relationships between attribute features and classes and provide a powerful means of expressing symbolic knowledge. We use a variational Expectation-Maximization (EM) algorithm to train the model in an end-to-end way and utilize the logic adapter to reasoning results during the test. Additionally, the logic adapter offers a reasoning process through FOLs, which enables the model to adapt from seen classes to recognize unseen classes and explain why a particular decision was made. Finally, we present the results of our experiments conducted on the AwA2 [25] and CUB [24] datasets to evaluate the performance and interpretability of ZSCLR. These results vividly illustrate the remarkable superiority and promising potential of ZSCLR in zero-shot image classification.

In summary, our contributions can be summarized as follows:

- To the best of our knowledge, ZSCLR is the first to integrate logic rules into zero-shot image classification. It includes a visual perception module and a logic adapter. It is a novel paradigm for zero-shot image classification.
- The visual perception module extracts basic image features via attributes guided, while the logic adapter enhances this process by utilizing rule prompts to attain discriminative image features. Furthermore, the logic adapter employs a Markov logic network to integrate rule prompts and the visual perception module and achieve reasoning. Importantly, our approach enables end-to-end training within a flexible variational EM framework. It not only enhances model performance for unseen classes but also provides interpretability for predictions, offering insights into the underlying reasoning process.
- Based on our extensive experimental results, we attain the superior performance of ZSCLR compared to the state-of-the-art methods. Furthermore, we illustrate the interpretability aspect of our model by offering visualizations that significantly enhance the comprehension of the underlying reasoning process.

## 2 RELATED WORK

Zero-shot learning aims to train a model to recognize classes not included in its original training. There are many works to study zero-shot learning in image classification. These works can be classified

into three categories: embedding-based approaches, generative-based approaches and knowledge-based approaches. Embedding-based approaches aim to learn a mapping function for visual-semantic interaction. They determine the label of a sample by matching their vectors in the same space using similarity metrics. Some approaches are implemented by mapping the visual features to semantic space by [17, 26, 28]. In contrast, other methods propose mapping the semantic features into visual space and point out that using semantic space as shared latent space may reduce the variance of features [34]. As the embedding is learned only on seen classes, these embedding-based methods inevitably overfit to seen classes. To address this problem, generative-based methods have been introduced utilizing generative models such as VAEs, GANs to learn semantic-visual mapping to generate visual features of unseen classes, which can offset the shortage of unseen classes and convert ZSL into a supervised classification task. This category focuses on learning a class condition generator to generate features of unseen classes [1, 32] or using semantic information (e.g., attributions) of class to generate features of unseen classes directly [5, 17, 21]. However, these methods still usually yield relatively undesirable results since they cannot capture the subtle differences between seen and unseen classes. Knowledge-based methods have been explored to capture more correlations between seen and unseen classes. Most approaches utilize knowledge graphs (e.g., class hierarchies and commonsense knowledge) as side information. Specifically, they are used to build relationship graphs between seen and unseen classes as a semantics-level graph to learn recognizing unseen classes [2, 10, 11, 23, 29].

Besides the three mentioned categories, the zero-shot image classification methods can also be divided into discriminative and non-discriminative methods based on whether they consider the importance of different features. Since the above-mentioned approaches fail to account for discriminative features, they are attributed to non-discriminative methods. Recently, some discriminative methods have begun to explore the discriminative image features using attention networks [3, 4, 6, 27, 36]. Xie et al. [27] constructs a region graph using parts of the object via the attention technique and achieves transferring knowledge between different classes. Chen et al. [3, 4] utilize mutually visual-attribute attention sub-net for semantic distillation, encouraging the model to explore the discriminative features for image. To solve the attribute imbalance and co-occurrence, Chen et al. [6] introduces an attribute-level contrastive learning mechanism.

To some extent, our model is a discriminative approach. In contrast to existing approaches, we incorporate logic rules as auxiliary information to capture discriminative image features and logical relationships between discriminative features and classes and employ the Markov logic network for prediction. Moreover, our model introduces rule prompts with symbolic discriminative features, providing interpretability compared to those who use attention networks to capture discriminative image features.

## 3 METHODOLOGY

**Problem Setting.** Zero-shot classification (ZSC) aims to recognize unseen classes by transferring knowledge from the seen domain $\mathcal{D}^s$ to the unseen domain $\mathcal{D}^u$. The training data for seen classes

is denoted as $\mathcal{D}^s = \{(x^s, y^s, A^s) | x^s \in \mathcal{X}^s, y^s \in \mathcal{Y}^s\}$, where $\mathcal{X}^s$ represents the image sets with class labels from $\mathcal{Y}^s$, and $A^s \in \mathbb{R}^{S \times m}$ represents the category attributes of the seen classes. Similarly, $\mathcal{D}^u = \{(x^u, y^u, A^u) | x^u \in \mathcal{X}^u, y^u \in \mathcal{Y}^u\}$ is data of the unseen classes. Additionally, $\mathcal{X} = \mathcal{X}^s \cup \mathcal{X}^u$. In ZSC, the class space is disjoint between the seen and unseen domains, i.e., $\mathcal{Y}^s \cap \mathcal{Y}^u = \varnothing$. The model is trained on the seen classes $\mathcal{Y}^s$ but is tested on the unseen classes $\mathcal{Y}^u$. To bridge the gap between seen and unseen categories, auxiliary information such as attribute descriptions $A^s$ and $A^u$ is required. To aid in understanding the paper, important notations and their descriptions have been provided in Table 1.

**Table 1: Important notations and their descriptions.**

| Notations | Descriptions |
|---|---|
| $\mathcal{X}^s, x^s$ | seen image |
| $\mathcal{X}^u, x^u$ | unseen image |
| $\mathcal{Y}^s, y^s$ | seen class label |
| $\mathcal{Y}^u, y^u$ | unseen class label |
| $a^s_y, A^s$ | seen class attribute |
| $a^u_y, A^u$ | unseen class attribute |
| $V, V'$ | image feature |
| $A_{tt}$ | attribute feature |
| $S$ | the number of seen classes |
| $\alpha$ | attribute weight matrix of image features |
| $\phi$ | prediction score |
| $R^s, R^u, R, r$ | logic rule (FOL) |
| $a_r$ | ground atom in a logic rule |
| $\mathcal{A}_r, \mathcal{A}$ | ground atom set(s) |
| $\varphi$ | potential function |
| $w$ | weight sets of the logic rules |
| $w_r$ | weight of a logic rule |

**Overview.** In Figure 2, ZSCLR consists of a visual perception module and a logic adapter. The visual perception module can learn a mapping function from visual space to semantic space to extract image features $V'$ for recognizing objects. In this paper, to train the visual perception module, we compute the inner product of both extracted image features and class attribute labels to attain a score of classification. To make extracted image features more discriminative, we design a logic adapter. The logic adapter initially receives image features from the visual perception module. It then models rule prompts through Markov logic networks to learn a joint probability distribution. In logic adapter, we compute posterior via feature network to predict attribute labels, and combine these attribute labels, e.g., $clon\_bill(V')$ according to rule prompts to infer class labels, e.g., $clay\_colored\_sparrow(V')$ through fuzzy logic reasoning. During training, this process can refine image features $V'$ via backpropagation to attain discriminative image features. In this process, the logic adapter serves a dual purpose: aiding the

**Figure 2: Overview of ZSCLR: ZSCLR comprises a visual perception module and a logic adapter. The visual perception module extracts image features using a CNN, fully connected layers (FCs), and an attention network guided by attribute features. The learned image features are then input into the logic adapter, which employs a Markov logic network to learn a joint probability distribution capturing shared variables between seen and unseen classes. This facilitates effective knowledge transfer in zero-shot image classification, allowing inference of attribute feature labels and class labels via feature network and fuzzy logic.**

model in acquiring discriminative image features and facilitating adaptation to new environments, i.e., recognizing unseen classes. In our ZSCLR, the input includes seen class images $\mathcal{X}^s$, corresponding attribute feature vectors $\boldsymbol{a}_y^s$, class attribute labels $\boldsymbol{A}^s$ and rule prompts $R^s$ of seen classes during training. In the testing phase, we input unseen class images $\mathcal{X}^u$, their corresponding attribute feature vectors $\boldsymbol{a}_y^u$, and the knowledge base $R^u$ containing the rule prompts for unseen classes.

### 3.1 Visual Perception Module

In the Visual Perception Module (VPM), we aim to map the visual features in visual space to the corresponding attribute features in the semantic space. Specifically, the VPM requires two inputs for obtaining basic image features $V'$: a set of images $\mathcal{X} = \{x_1, ..., x_N\}$, where $N$ is the number of images, and matrix of the attribute features $A_{tt} \in \mathbb{R}^{N \times m}$, where $m$ is dimension of attribute feature. The initial features, represented by $V \in \mathbb{R}^{N \times m}$, are obtained by feeding these into a CNN, and then the fully connected layers with ReLU activation. As previous work [13] claims that learning useful visual features is crucial in ZSC, we coarsely screen out good image features as basic image features, denoted as $V' \in \mathbb{R}^{N \times m}$, using the attention network guided by attribute features from a given image. These basic image features are prepared for the logic adapter to refine the discriminative image features. Specifically, we need to calculate the attention for each feature by

$$\boldsymbol{\alpha} = \text{softmax}(A_{tt} W_a V), \qquad (1)$$

where $\boldsymbol{\alpha} \in \mathbb{R}^{N \times m}$ represents attribute weight matrix of image features, and $W_a \in \mathbb{R}^{m \times N}$ is a learnable matrix.

Next, initial visual features $V$ and $\boldsymbol{\alpha}$ calculate Hadama product to attain new visual features $V'$ by

$$V' = \boldsymbol{\alpha} \odot V, \qquad (2)$$

where $\odot$ is the Hadama product.

Then, $V'$ are refined and updated by the logic adapter in Section 3.2. Finally, we calculate similar scores for both basic image features and class attribute labels to attain image class labels. Note, this score calculation is performed solely during training to generate the loss Eq. (6) in VPM, thereby facilitating model training. More concretely, basic image feature $V'$ match its corresponding seen class attribute labels $A^s$, which is formulated as compute inner product:

$$\boldsymbol{\phi} = V' A^{s\top}, \qquad (3)$$

where $\boldsymbol{\phi} \in \mathbb{R}^{N \times S}$ is the score that represents the distribution of classes for the images.

### 3.2 Logic Adapter

As the core of ZSCLR, the logic adapter (LA) integrates the VPM with rule prompts to extract discriminative image features. Meanwhile, LA can adapt the trained model from seen to unseen classes, and enables reasoning through Markov logic networks. To achieve this integration, it's crucial to establish a unified representation between the VPM and rule prompts. Thus, we introduce Markov logic networks (MLN), a framework that seamlessly combines statistical models (VPM) with relational models (rule prompt) into a unified representation [7]. For this purpose, we use FOLs like dagger_bill($x$) ∧ brown_wing($x$) ⇒ brewer_sparrow($x$) as rule

prompts. These FOLs establish logical relationships between symbolic discriminative features (attributes) and classes. Specifically, LA uses MLN to learn a joint probability distribution for symbolic discriminative features, enabling the prediction of discriminative feature labels by calculating posterior probabilities. The class labels are then inferred using fuzzy logic based on the predicted discriminative feature labels.

In fact, MLN can be thought of as a knowledge base utilizing FOLs. In MLN, we assume there is a FOL set $R$ for a given dataset, and an FOL $r \in R$ comprises a set of predicate functions, like $p_r^1(x)$, $p_r^2(x)$, etc., where the domain of $x$ is the set of constants $C = \{c_1, c_2, ...\}$, $c_i$ denotes object $i$. When the constants are assigned to the arguments of a predicate, these assigned predicates are called `ground atoms`, and each FOL corresponds to a ground atom set $\mathcal{A}_r = \{a_r^1, a_r^2, ...\}$. Using the examples from Figure 1 (a), the constants $c_1$ and $c_2$ are the image features $V_1'$ and $V_2'$ obtained by the VPM, the ground atom sets are $\mathcal{A}_1 = \{$`cone_bill`$(V_1')$, `grey_wing`$(V_1')$, `clay_colored_sparrow`$(V_1')$, `cone_bill`$(V_2')$, `grey_wing`$(V_2')$, `clay_colored_sparrow`$(V_2')\}$, and $\mathcal{A}_2 = \{$`dagger_bill`$(V_1')$, `brown_wing`$(V_1')$, `brewer_sparrow`$(V_1')$, `dagger_bill`$(V_2')$, `brown_wing`$(V_2')$, `brewer_sparrow`$(V_2')\}$ for the two FOLs, i.e., the rule prompts.

MLN models FOLs as a probabilistic graph, where nodes represent ground atoms, and edges between nodes correspond to logical relationships between these ground atoms. For the FOL set $R$ and its ground atoms, the joint distribution defined by MLN can be represented as follows:

$$P_w = \frac{1}{Z(w)} \exp\{\sum_{r \in R} w_r \sum_{a_r \in \mathcal{A}_r} \varphi(a_r)\}, \tag{4}$$

where $Z(w)$ is the partition function summing overall ground atoms. $w$ represents the weight sets of all rules, and $w_r$ is the corresponding weight of each FOL. $\varphi$ denotes a potential function in terms of the number of times the logic rule is true.

## 3.3 Model Optimization

We consider both the VPM and LA in our model and optimize them end-to-end. For the VPM, we follow the same way as in LFGAA [13] that uses triplet loss [18] to learn image features by simultaneously enlarging interclass distance and reducing intra-class distance:

$$L_T = \frac{1}{N} \sum_{i=1}^{N} [||V_i' - V_j'||^2 - ||V_i' - V_m'||^2 + \beta]_+, \tag{5}$$

where $V_i'$, $V_j'$ and $V_m'$ serve as anchor, positive and negative sample within a triplet respectively. $[\circ]_+$ is equivalent to $max(\circ, 0)$. $\beta$ is a hyperparameter controlling the desired margin between the positive and negative pairs.

Furthermore, to learn a mapping function from visual features to semantic space, we design the following loss:

$$L_S = -\frac{1}{N} \sum_{i=1}^{N} \log \frac{\exp(\phi_i)}{\sum_S \exp(\phi_i)}, \tag{6}$$

where $\phi_i$ represents class label score of each image.

Therefore, the VPM's final loss can be written as follows:

$$L_V = L_T + L_S. \tag{7}$$

Logic adapter is a MLN, we need to maximize the log-likelihood of all the observed predicates (variables) $\log P_w$. However, it is intractable to maximize the overall objective directly since it requires computing the whole partition function $Z(w)$ and integrating over all predicates. Therefore, as suggested in [31], the approach is to optimize the variational evidence lower bound (ELBO) of the data log-likelihood, which is as follows:

$$L_{ELBO} = E_Q[\log P_w] - E_Q[\log Q(\mathcal{A})], \tag{8}$$

where $\mathcal{A}$ is ground atom sets. $Q(\mathcal{A})$ is the variational posterior.

We use the variational EM algorithm to optimize $L_{ELBO}$, i.e., to minimize KL divergence between the $Q(\mathcal{A})$ and $P_w$ to implement inference during E-step. In the M-step, the algorithm uses gradient descent to learn the weight of the FOL. Due to the inference of the MLN is #$P$-complete [16], we need to use mean-field theory to factorize variational distribution $Q(\mathcal{A})$ to approximate the $P_w$:

$$Q(\mathcal{A}) = \prod_{\mathcal{A}_i \in \mathcal{A}} Q(\mathcal{A}_i). \tag{9}$$

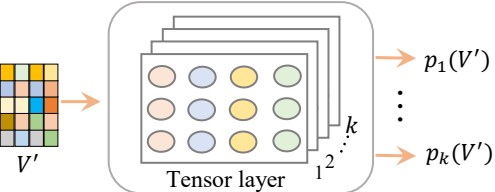

Feature network

**Figure 3: Feature network. The inputs are image features of the object, e.g., $V'$, and the outputs are probabilities of the attribute labels of the object, e.g., $p_k(V')$. $k$ represents the tensor layer, and each layer is a predicate.**

To improve inference effectiveness, we use neural networks (feature networks in this paper) to parameterize variational calculation in Eq. (9). Consequently, this variational operation becomes a process of learning parameters. In this paper, we use a tensor network as our feature network to model $Q$ as $Q_\theta$ in Figure 3, and $\theta$ is the parameter of the feature network.

Thus, according to Eq. (4), Eq. (9) and feature network, the Eq. (8) can be rewritten as follows:

$$L_{ELBO} = \sum_{r \in R} w_r \sum_{a_r \in \mathcal{A}_r} E_{Q_\theta}[\varphi(a_r)] - \log Z(w)$$
$$- \sum_{\mathcal{A}_i \in \mathcal{A}} E_{Q_\theta}[\log Q_\theta(\mathcal{A}_i)]. \tag{10}$$

Finally, we combine optimization targets of the visual perception module and logic adapter to minimize final loss $L$:

$$L = L_V - L_{ELBO}. \tag{11}$$

In the M-step, the model needs to learn the weights of the FOLs. The partition function $Z(w)$ has an exponential number of terms, which makes it intractable to optimize ELBO directly. To address this issue, we use pseudo-likelihood [16] to optimize $L_{ELBO}$ in M-step, as the following:

$$P_w^* := E_{Q_\theta}[\sum_{r, \mathcal{A}_i \in \mathcal{A}_r} \log P_w(\mathcal{A}_i | MB_{\mathcal{A}_i})], \tag{12}$$

where $MB_{\mathcal{A}_i}$ is Markov blanket of the ground atom $\mathcal{A}_i$. For each rule $r$ that connects $\mathcal{A}_i$ to its Markov blanket, we optimize the weights $w_r$ by gradient descent, the derivative is following:

$$\nabla_{w_r} E_{Q_\theta}[\log P_w(\mathcal{A}_i|MB_{\mathcal{A}_i})] \simeq Y_{\mathcal{A}_i} - P_w(\mathcal{A}_i|MB_{\mathcal{A}_i}), \quad (13)$$

where $Y_{\mathcal{A}_i} = 0$ or 1 if $\mathcal{A}_i$ is an observed variable, and $Y_{\mathcal{A}_i} = Q_\theta(\mathcal{A}_i)$ otherwise.

### 3.4 Inference

**Prediction**. After training ZSCLR, the model recognizes unseen images using the logic adapter. Specifically, the model uses the visual perception module to extract the discriminative image features, denoted as $V'$. These discriminative image features are then input into the logic adapter. In the logic adapter, the feature network is used to calculate the posterior probability $Q(\mathcal{A}_i)$, attaining the attribute feature labels of the instances. Then, the model leverages rule prompts from unseen classes to combine recognized attribute feature labels to infer class labels via fuzzy logic reasoning.

**Interpretability**. Our ZSCLR offers interpretability for predictions through corresponding rule prompts. That is to say, the model not only can predict class labels of objects in the images but also tell reasons for classifying the images to the class labels through symbolic language. In this study, we employ key attribute characteristics as our discriminative image features. To enhance understanding, we represent these attribute features using symbolic logic rules, replacing them in vector form. For example, when the model recognizes an unseen class image, it can identify the key attribute characteristics present in the sample and subsequently classify the unseen class object based on both these key attribute characteristics and the associated rule prompts. In this process, the key attribute characteristics and their corresponding rule prompts serve as explanations for the predictions.

## 4 EXPERIMENT

### 4.1 Experimental Setup

**Datasets.** We employed two challenging benchmark datasets, AwA2 (Animals with Attributes 2) [25] and CUB (Caltech UCSD Birds 200-2011) [24], to validate our method. These datasets offer varying degrees of granularity, with AwA2 being a coarse-grained dataset and CUB being fine-grained. AwA2 consists of 37,322 images distributed across 50 animal categories, each associated with 85 attributes. Within this dataset, 40 categories are considered seen during training, while the remaining 10 are unseen during training and are used for evaluation. CUB, on the other hand, comprises 11,788 images spanning 200 bird classes, each associated with 312 attributes. Among these classes, 150 classes are designated as seen during training, while the remaining 50 are unseen and used for evaluation. Moreover, to enable logic-based reasoning, we constructed logic rules based on attributes, taking the form of $\texttt{attribute}_1(x) \wedge \texttt{attribute}_2(x) \wedge \cdots \wedge \texttt{attribute}_n(x) \Rightarrow \texttt{class}(x)$. In these rules, the rule body consists of attributes, while the rule head represents class labels. To divide the classes into seen and unseen categories, we adopted the Proposed Split (PS) method [25]. The statistics of the datasets are shown in Table 2.

**Evaluation Protocols.** We followed the evaluation protocol outlined in [25] and assessed the top-1 accuracy in two distinct settings.

**Table 2: Dataset statistics.**

| Datasets | Attributes | $|\mathcal{Y}^s|$ | $|\mathcal{Y}^u|$ | $|X^s|$ | $|X^u|$ | $|R|$ |
|---|---|---|---|---|---|---|
| AwA2 | 85 | 40 | 10 | 30,414 | 6,908 | 85 |
| CUB | 312 | 150 | 50 | 7,057 | 4,731 | 312 |

First, we conducted experiments within the conventional zero-shot learning (CZSL) framework, where exclusively unseen categories were involved in testing. Consequently, all prediction results were restricted to be drawn from unseen classes, and this accuracy is denoted as Acc. Secondly, we ventured into the realm of generalized zero-shot learning (GZSL), wherein test images encompassed both seen and unseen categories. In this setting, we computed the top-1 accuracy for both seen (S) and unseen (U) categories independently. Furthermore, to gauge the trade-off between performance on seen and unseen categories, we calculated the harmonic mean (H) by using the formula $H = 2 \times \frac{S \times U}{S + U}$.

**Implementation Details.** We employed CNN as our basic image features extractor in the visual perception module. Indeed, our CNN can be initialized by a pre-trained backbone such as GoogleNet [20]. Before being fed into the model, the images need to be randomly cropped for data augmentation. For optimization, we used the Adam optimizer with specific configurations. On the AwA2 dataset, we set the number of epochs to 15, the batch size to 64, and the learning rate to 1e-4. On the CUB dataset, we conducted training for 20 epochs, with a batch size of 32 and a learning rate of 1e-5.

**Baselines.** We compare our ZSCLR with representative approaches proposed in recent years. These approaches are divided into two classes: No-Discriminative methods include E-PGN [32], Composer [8], GCM-CF [33], FREE [5], LFGAA [13], DAZLE [9], APN [28], CF-ZSL[29]; Discriminative methods include MSDN [4], TransZero [3], DUET[6].

### 4.2 Comparision with State-of-the-Arts

**Conventional Zero-Shot Learning.** Table 3 presents the results of CZSL on various datasets. Our ZSCLR achieves competitive Acc results on AwA2 and CUB, respectively. This shows that ZSCLR captures the discriminative visual representations for distinguishing unseen classes and utilizes the logic relationship between seen and unseen classifications via MLN. Comparing results on AwA2, our ZSCLR performs better on CUB. This is because AwA2 is a coarse-grained dataset, and attribute semantics are relatively abstract, e.g., big or small, while CUB is a fine-grained dataset, and its attribute semantics are specific, e.g., shape of bill. Based on these fine-grained object contour semantics, ZSCLR can learn decent image features on CUB and transfer them to unseen classes via fine-grained logic rules. Moreover, we observed that discriminative methods outperform non-discriminative methods in most cases. This validates the importance of discriminative features in recognizing objects.

**Generalized Zero-Shot Learning.** Table 3 also presents the results of different methods in the GZSL setting. It is evident that many methods achieve good results on seen classes but struggle with unseen classes in the CUB and AWA2 datasets. In contrast, our ZSCLR performs well in unseen classes. This advantage can be attributed to the logic adapter in ZSCLR, which enables the capture of discriminative image features for effective knowledge transfer from seen to unseen categories.

**Table 3: Results (%) of our method and compared baselines. The best results in baselines are marked in bold. "-" is not reported in their paper.**

| Classes | Methods | AWA2 | | | | CUB | | | |
|---|---|---|---|---|---|---|---|---|---|
| | | CZSL | GZSL | | | CZSL | GZSL | | |
| | | Acc | U | S | H | Acc | U | S | H |
| No-Discriminative | E-PGN [32] | 73.4 | 52.6 | 83.5 | 64.6 | 72.4 | 52.0 | 61.1 | 56.2 |
| | Composer [8] | 71.5 | 62.1 | 77.3 | 68.8 | 69.4 | 56.4 | 63.8 | 59.9 |
| | GCM-CF [33] | - | 60.4 | 75.1 | 67.0 | - | 61.0 | 59.7 | 60.3 |
| | FREE [5] | - | 60.4 | 75.4 | 67.1 | - | 55.7 | 59.9 | 57.7 |
| | LFGAA [13] | 68.1 | 27.0 | **93.4** | 41.9 | 67.6 | 36.2 | **80.9** | 50.0 |
| | DAZLE [9] | 67.9 | 60.3 | 75.7 | 67.1 | 66.0 | 56.7 | 59.6 | 58.1 |
| | APN [28] | 68.4 | 57.1 | 72.4 | 63.9 | 72.0 | 65.3 | 69.3 | 67.2 |
| | CF-ZSL[29] | 69.2 | 33.3 | 82.0 | 47.4 | 66.2 | 36.3 | 72.9 | 48.5 |
| Discriminative | MSDN [4] | 70.1 | 62.0 | 74.5 | 67.7 | 76.1 | 68.7 | 67.5 | 68.1 |
| | TransZero [3] | 70.1 | 61.3 | 82.3 | 70.2 | 76.8 | 69.3 | 68.3 | 68.8 |
| | DUET[6] | 69.9 | 63.7 | 84.7 | 72.7 | 72.3 | 62.9 | 72.8 | 67.5 |
| | ZSCLR (our) | **76.0** | **70.4** | 76.7 | **73.4** | **77.8** | **70.5** | 74.3 | **72.4** |

**Table 4: Ablation studies for different compositions of ZSCLR. The *Att Net* is the attention network.**

| Methods | AwA2 | CUB |
|---|---|---|
| VPM w/o *Att Net* | 62.2 | 51.0 |
| VPM w/ *Att Net* | 68.4 | 67.6 |
| VPM w/o *Att Net*, w/ LA | 70.3 | 70.5 |
| ZSCLR | 76.0 | 77.8 |

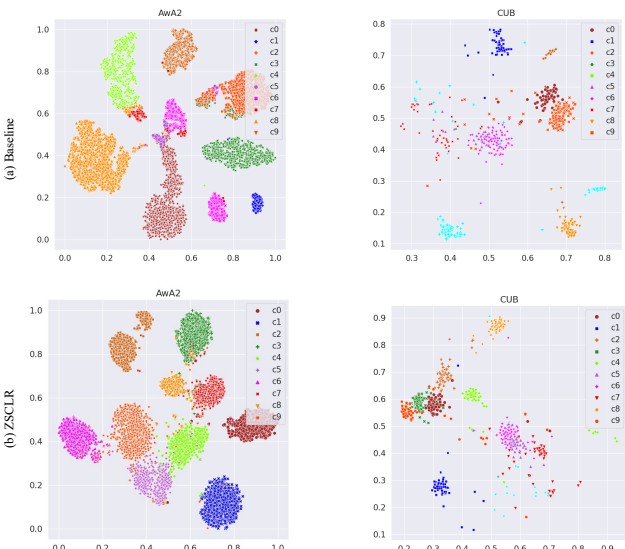

**Figure 4: t-SNE visualizations of visual features for unseen classes on AwA2 and CUB dataset in CZSL, respectively. (a) represents the baseline method, while (b) is our ZSCLR. $C_i$ represents different unseen classes.**

## 4.3 Ablation Studies

We evaluate our model to compare performance gain (Acc) brought by different components on the AwA2 and CUB datasets as shown in Table 4. Specifically, we first observe a sharp drop in performance if using only a visual perception module with CNN (VPM w/o *Att Net*) to predict unseen class images via computing the inner product of both extracted image features and class attribute labels. Second,

from the results obtained using a visual perception module with both CNN and attention network as the image feature extractor to predict results through the inner product of extracted image features and class attribute labels (VPM w/ *Att Net*). We observe that employing an attribute-guided strategy for extracting visual features benefits our model. Next, we also assess the influence of the logic adapter, where the visual perception module with CNN and predict results by logic adapter (VPM w/o *Att Net*, w/ LA), which results in better performance compared to the VPM w/ *Att Net*. This demonstrates that the logic adapter not only benefits the capture of discriminative image features but also adapts the model to unseen classes. Finally, we find that by combining all these components, the model can achieve the best results.

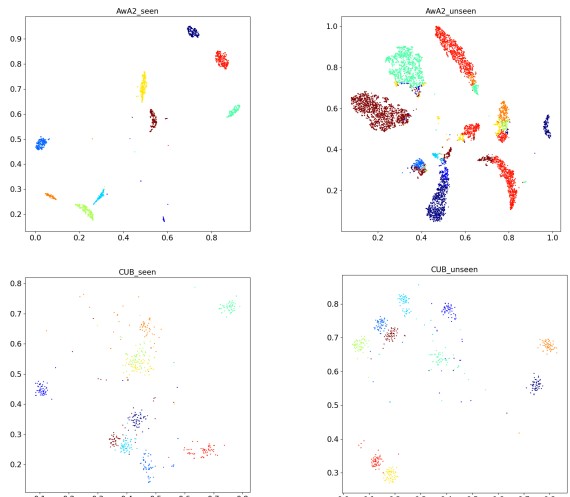

**Figure 5: t-SNE visualizations of visual features for both seen and unseen classes on AwA2 and CUB datasets in GZSL, respectively. We show 10 classes in this experiment.**

## 4.4 Qualitative Results

**Discriminative Image Features Analyze**. *In CZSL scenes*. To demonstrate ZSCLR's ability to capture discriminative image features, we visualize features for unseen classes using t-SNE on the

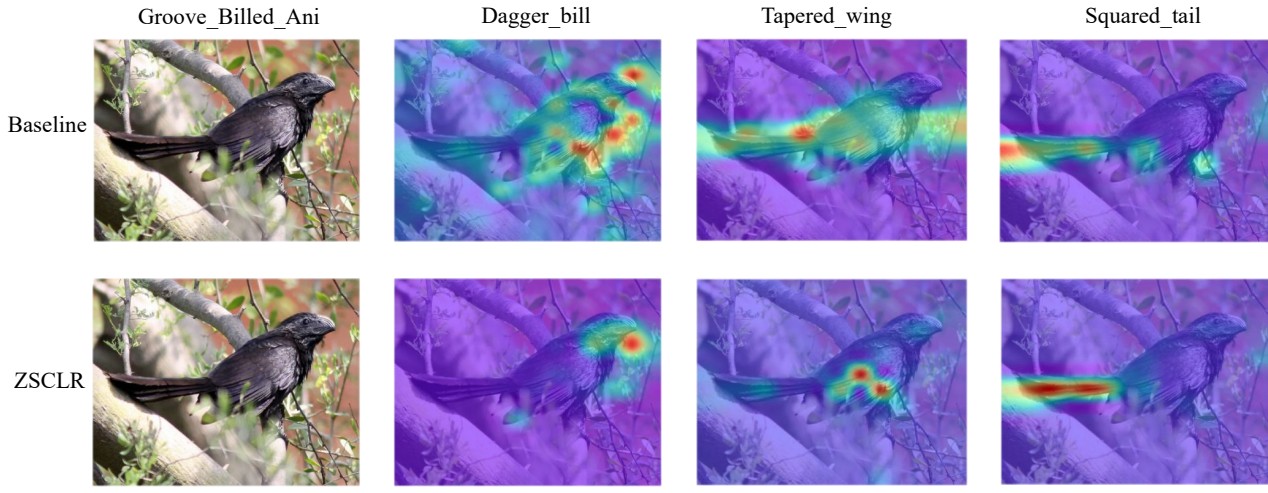

Rule prompt: $dagger\_bill(x) \wedge tapered\_wing(x) \wedge squared\_tail(x) \implies groove\_billed\_ani(x)$

**Figure 6: Interpretabilty. Visualization of learned discriminative image features for baseline (LFGAA [13]) and our ZSCLR on the CUB dataset. We highlight the position of three discriminative features, such as the shape of the bill, the shape of the wing, and the shape of the tail in an image.**

AwA2 and CUB datasets (Figure 4). Compared to the baseline [13], ZSCLR displays distinct clusters. This affirms that our model effectively learns discriminative and transferable features. It also underscores the logic adapter's role in prompting the visual perception module to capture fine-grained attribute semantics shared between seen and unseen classes, resulting in the learning of discriminative visual representations that facilitate knowledge transfer. *In GZSL scenes*. We visualize visual features of both seen and unseen classes in the AwA2 and CUB datasets in Figure 5. To facilitate visualization, we randomly selected 10 test classes for each dataset. We observed that our ZSCLR shows better clustering results on both seen and unseen classes. This demonstrates the superiority and potential of ZSCLR for knowledge transfer.

**Effectiveness and Interpretability Analyze**. To provide a clear illustration of ZSCLR's effectiveness and interpretability, we have employed heatmaps to visualize the discriminative image features learned by the proposed model ZSCLR and a baseline (LFGAA [13] as an example) without logic knowledge on the CUB testing data. As depicted in Figure 6, the highlighted regions represent the captured discriminative features. Compared to the baseline, our ZSCLR obtains better results, e.g., accurately positioning by paying attention to discriminative feature regions. This demonstrates ZSCLR's effectiveness in capturing discriminative image attributes utilizing rule prompt knowledge. Furthermore, by combining the predicted discriminative feature labels with the rule prompts, ZSCLR can infer class labels, e.g., *groove_billed_ani* and provide an interpretation of the results. This reasoning process is transparent, allowing for easy comprehension of the model's recognition process when presented with an image. For instance, when the model identifies an image as *groove_billed_ani*, it can provide an explanation by highlighting the presence of features like *digger_bill*,

*tapered_wing* and *squared_tail* in the image, and then logically deducing that the object possessing these features corresponds to the *groove_billed_ani* class, according to the provided rule prompts.

## 5 CONCLUSION AND DISCUSSION

In this paper, we propose the zero-shot image classification model with logic adapter and rule prompt (ZSCLR) to more accurately classify images in zero-shot scenes and provide the results' interpretability. ZSCLR consists of two modules: a visual perception module and a logic adapter. The visual perception module aims to extract basic image features. At the same time, the logic adapter takes basic image features from the visual perception module and encodes them and rule prompts via the Markov logic network. During training, the logic adapter refines these basic image features using backpropagation to derive discriminative image features and adapt the model from seen classes to unseen classes. Furthermore, ZSCLR offers explanations for its prediction results through rule prompts with symbolic discriminative features. Comprehensive experiments conducted on two well-known benchmarks underscore the superior performance of ZSCLR. We believe that ZSCLR presents an innovative direction for the research community, particularly in the context of logic reasoning.

Our ZSCLR not only predicts classes of images but also explains reasons for predictions by providing an interpretation in natural language form. Meantime, the model can locate the position of the discriminative features of objects. Therefore, in the future, ZSCLR has the potential to integrate into visual question models with natural language interactive Q&A. For instance, given an image and its prompt like "there are *bill*, *wing* and *tail* in the image" in the natural language style, the model can answer the following questions like "*What is an object in this image?*", "*Why is the object?*", "*Where are the discriminative features for determining the object?*", and so on.

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
