# OpenReview forum: "Zero-shot Image Classification with Logic Adapter and Rule Prompt"
_ACM.org/TheWebConf/2024/Conference — TheWebConf24_

### Official Review · Reviewer_BFhf · 2023-11-23

**Novelty:** 3
**Technical Quality:** 3

**Review:**

Strength:
This paper proposes to integrate logic rules into zero-shot image classification. The proposed method enables end-to-end training within a flexible variational EM framework. ZSCLR achieves superior performance compared to state-of-the-art methods.

Weakness:
1. The relevance of this work to the conference track should be clarified.
2. Figure styles: The styles of Figures 2 and 3 need to be more unified. Their color scheme is overly complex and confusing. Additionally, Figure 2 contains too much text, which hinders readability. Simplifying the decorative style in these drawings will improve their clarity.
3. Presentation: It is better to place tables and figures at the top of a page and close to their text descriptions. Section 3 is not concise and clear enough and needs to be read many times while scrolling to Table 1 to find the meaning of the keywords. Make sure terms like "pr1" and "pr2" (mentioned in line 475) are clearly defined. Are they linear projections?
4. Analysis in Table 3: Analysis of the performance differences as shown in Table 3 should be provided, specifically why ZSCLR performs significantly lower than GZSL(S) and other methods. This insight is currently lacking but is important for understanding the context of the results.
5. Ablation Study: The paper currently only contains one Ablation Study, which is not enough. Expanding this section with additional ablation experiments will enhance the validity of the claimed findings.

Overall, careful revision of the figures, text, and structure will significantly improve its quality and readability

**Questions:**

1. Please analyze the performance differences noted in Table 3, specifically why ZSCLR performs significantly lower than GZSL(S) and other methods. This insight is currently lacking but is important for understanding the context of the results.
2. The paper currently only contains one ablation study, which is insufficient. Expanding this section with additional experimental analysis will enhance the validity of the findings.

**Reviewer Confidence:**

3: The reviewer is confident but not certain that the evaluation is correct

**Scope:**

3: The work is somewhat relevant to the Web and to the track, and is of narrow interest to a sub-community

---

### Official Review · Reviewer_jGpa · 2023-11-23

**Novelty:** 6
**Technical Quality:** 5

**Review:**

The approach proposes a novel method to use logic rules to improve the zero-shot image classification.

Strengths:
- The approach obtains state-of-the-art performance
- The approach allows focusing on discriminative features

Weaknesses:
- The approach relies on FOL rules that have to be manually created and might be difficult to create and in some cases impossibles

**Questions:**

- What types of FOL rules does the approach support and how complex?

- What happens if the image is a partial image and some important features are not in the image? Could the approach identify correctly the label?

**Reviewer Confidence:**

3: The reviewer is confident but not certain that the evaluation is correct

**Scope:**

3: The work is somewhat relevant to the Web and to the track, and is of narrow interest to a sub-community

---

### Official Review · Reviewer_BEaw · 2023-11-24

**Novelty:** 4
**Technical Quality:** 4

**Review:**

The paper proposed a new model for zero-shot learning called ZSCLR. It is the first model integrate logic rules, zero-shot image classification, Markov logistic network and EM algorithm. The feature of the image is first passed through a visual perception module which is composed of CNN and attention networks. One of the novelty of the paper is the logic adapted which extract the features of the image via attributes and generate rule prompts to refine the latency with the help of evidence lower bound of the data log-likelihood while being optimized.

Pro:
1.The paper proposed a new zero-shot learning algorithm which combines the advantage of both traditional CNN-Attention mixed network and Markovian logistic network, which extracts the discriminative features of the object and turns them into prompts to refine the original feature produced by the Vision Perception Module
2.The algorithm outperforms greatly the state-of-the-art methods on two classical zero-shot learning datasets and provides abundant ablation studies while making clear visualization charts.
3.The structure of the paper is clear. The methodology is split into several parts which corresponds to different modules of the whole model. The logistic adapter added into traditional Vision Perception Networks shows the novelty of the work.
4.The illustration is adequate and explicit enough for the readers and researchers to follow up with the idea of the author.

Con:
1.The experiments are constructed on only two datasets: Animals with Attributes and Caltech UCSD Birds. More experiments are encouraged to be conducted.
2.The paper lacks original theoretical deduction. More analysis of the increasing accuracy of the performance is encouraged to explain the availability of the proposed algorithm.

**Questions:**

1.Can the logistic adapter be replaced with other variants or methods, such as diffusion-like model or some other probability model?
2.Will the model depicted in the article show great performance on other datasets? The experiments part only gives out the result of two datasets which may not be convincible enough.

**Reviewer Confidence:**

3: The reviewer is confident but not certain that the evaluation is correct

**Scope:**

3: The work is somewhat relevant to the Web and to the track, and is of narrow interest to a sub-community

---

### Official Review · Reviewer_n2vQ · 2023-11-27

**Novelty:** 6
**Technical Quality:** 6

**Review:**

The paper introduces ZSCLR, a method which combines a visual perception module and a logic adapter for the task of Zero Shot Image Classification.

Pros
* novelty of the proposed approach
* better performances wrt soa
* interpretabili feature

Cons
* lack some clarity (see questions) which may lead to even more needed annotations

**Questions:**

* It isn't clear if the attribute labels should be further annotated, also for the unseen image. In this case, all attributes should be "seen"?
* Sec 4.1. "We constructed" . This means that you need a dataset annotated with these prompts? How did you realise this? How many annotators?
* Sec 4.4 "better clustering" than what? Actually all this section is not convincing, not sure if the figures are correct but I see good clusters in the baseline too. You may want to have also a quantitative metric for the clustering quality

**Ethics Review Description:**

N.a.

**Reviewer Confidence:**

2: The reviewer is willing to defend the evaluation, but it is likely that the reviewer did not understand parts of the paper

**Scope:**

3: The work is somewhat relevant to the Web and to the track, and is of narrow interest to a sub-community

---

### Decision · Program_Chairs · 2024-01-22

**Decision:**

Accept

**Comment:**

The paper presents ZSCLR, a novel method combing a visual perception module and a logic adapter for Zero Shot Image Classification. This model stands out as the first to integrate logic rules, zero-shot image classification, Markov logistic networks, and the EM algorithm. The image's features undergo processing through a visual perception module consisting of CNN and attention networks. A key innovation lies in the logic adapter, which extracts image features using attributes and generates rule prompts to enhance latency refinement, optimizing the evidence of the data log-likelihood lower bound.

 The paper fits to the scope of the Semantics track. The reviewers consider the paper to be a novel contribution of fair significance. The technical quality of the paper is also considered adaequate. The authors provide additional experimental results in the rebuttal.

 Pros:
 1. The paper exhibits clear structure and sufficient illustrations for readers to grasp the author's ideas effectively.
 2. It introduces a pioneering zero-shot learning algorithm, merging the strengths of traditional CNN-Attention mixed networks and Markovian logistic networks.
 3. Furthermore, it pioneers the integration of logic rules into zero-shot image classification, enabling a focus on discriminative features and enhancing interpretability.

 Cons:
 1. The paper lacks original theoretical deduction
 2. Evaluation experiments are considered limited since they are restricted to two datasets only
 3. The approach relies on FOL rules that have to be manually created and might be difficult to create and in some cases impossibles